# Spontaneous Ventilation Thoracoscopic Lung Biopsy in Undetermined Interstitial Lung Disease: Systematic Review and Meta-Analysis

**DOI:** 10.3390/jcm13020374

**Published:** 2024-01-10

**Authors:** Alexandro Patirelis, Stefano Elia, Benedetto Cristino, Ermanno Puxeddu, Francesco Cavalli, Paola Rogliani, Eugenio Pompeo

**Affiliations:** 1Unit of Thoracic Surgery, Department of Surgical Sciences, University of Rome Tor Vergata, 00133 Rome, Italy; alexandro.patirelis@hotmail.it (A.P.); crsbdt00@uniroma2.it (B.C.); 2Division of Thoracic Surgery, University Hospital Policlinico Tor Vergata, 00133 Rome, Italy; stefano.elia@unimol.it; 3Department of Medicine and Health Sciences, Università degli Studi del Molise, 86100 Campobasso, Italy; 4Unit of Respiratory Medicine, Department of Experimental Medicine, University of Rome Tor Vergata, 00133 Rome, Italy; ermanno.puxeddu@uniroma2.it (E.P.); roglianp@uniroma2.it (P.R.); 5Division of Respiratory Medicine, University Hospital Policlinico Tor Vergata, 00133 Rome, Italy; francesco.cavalli@ptvonline.it

**Keywords:** interstitial lung disease, surgical biopsy, non-intubated thoracic surgery, awake thoracic surgery, spontaneous ventilation, thoracoscopic surgery, VATS, idiopathic pulmonary fibrosis

## Abstract

Thoracoscopic surgical biopsy has shown excellent histological characterization of undetermined interstitial lung diseases, although the morbidity rates reported are not negligible. In delicate patients, interstitial lung disease and restrictive ventilatory impairment morbidity are thought to be due at least in part to tracheal intubation with single-lung mechanical ventilation; therefore, spontaneous ventilation thoracoscopic lung biopsy (SVTLB) has been proposed as a potentially less invasive surgical option. This systematic review summarizes the results of SVTLB, focusing on diagnostic yield and operative morbidity. A systematic search for original studies regarding SVTLB published between 2010 to 2023 was performed. In addition, articles comparing SVTLB to mechanical ventilation thoracoscopic lung biopsy (MVTLB) were selected for a meta-analysis. Overall, 13 studies (two before 2017 and eleven between 2018 and 2023) entailing 675 patients were included. Diagnostic yield ranged from 84.6% to 100%. There were 64 (9.5%) complications, most of which were minor. There was no 30-day operative mortality. When comparing SVTLB to MVTLB, the former group showed a significantly lower risk of complications (*p* < 0.001), whereas no differences were found in diagnostic accuracy. The results of this review suggest that SVTLB is being increasingly adopted worldwide and has proven to be a safe procedure with excellent diagnostic accuracy.

## 1. Introduction

Interstitial lung diseases (ILDs) are defined as a heterogeneous group of idiopathic or secondary conditions that share a similar clinical, radiological, and histologic pattern characterized by inflammation and fibrosis involving the lung interstitium [1]. Overall, the prevalence of ILDs is approximately 200 individuals per 100,000 [2]. An early and accurate differential diagnosis among undetermined ILDs is usually pivotal to predicting prognosis and determining the best pharmacological therapy. International consensus guidelines recommend the need for a precise histopathological diagnosis in undetermined ILD patients in whom clinical, laboratory, and high-resolution computed tomography findings, preferably discussed within a multidisciplinary panel, prove insufficient to make a confident diagnosis [2,3]. In these instances, surgical lung biopsy is still the recommended diagnostic procedure despite the proposal of alternative techniques such as transbronchial lung cryobiopsy [4]. Nowadays, mechanical ventilation thoracoscopic lung biopsy (MVTLB), performed under general anesthesia with double-lumen tube intubation, is the preferred surgical diagnostic tool, offering excellent diagnostic yields of 93–95% [5,6]. However, given the typical impaired pulmonary function of ILD patients, the risk of post-operative complications of MVTLB is still not negligible, with reported mortality and morbidity rates of 1.5–2.4% and approximately 16%, respectively [6,7,8,9]. In addition, general anesthesia and single-lung mechanical ventilation entail multiple potential side effects, which are thought to be a risk factor for the acute exacerbation of ILD [7]. In an attempt to combine the excellent diagnostic yield of MVTLB with lower operative risks, spontaneous ventilation thoracoscopic lung biopsy (SVTLB), performed under regional anesthesia without tracheal intubation and single-lung mechanical ventilation, has been proposed as a less invasive surgical alternative [10]. This study aims to review all cohort studies on SVTLB for ILD and evaluate their outcomes in terms of safety and diagnostic accuracy.

## 2. Materials and Methods

### 2.1. Literature Search

A systematic search of 2 databases (PubMed and Google Scholar) was performed in September 2023, considering the publication period from January 2010 to August 2023. The used search terms are hereafter reported: “interstitial lung disease” or “idiopathic pulmonary fibrosis” in combination with “video-assisted thoracic surgery” or “VATS” or “thoracoscopic” or “surgical biopsy” and with “non-intubated” or “awake” or “regional anesthesia”. The search was restricted to human adult (≥18 years old) patients and to English studies. Bibliographies of selected articles were reviewed for additional publications.

### 2.2. Outcome Definition

For the purpose of this study, the 2 main outcomes included in the systematic review were post-operative morbidity and diagnostic yield. Articles comparing SVTLB versus MVTLB were considered for a subgroup meta-analysis.

### 2.3. Study Selection

All articles found in the research databases were independently reviewed by two authors (EP and AP), sequentially evaluating the title, the abstract, and the full text. The inclusion criteria for eligibility were as follows: (1) original research; (2) patients over 18 years of age and with undetermined ILD; (3) surgical biopsy originally performed with spontaneous ventilation/awake anesthesia protocols; (4) a minimum of 10 enrolled patients; (5) report of at least one main outcome; (6) English language. When multiple publications from the same authors were present within an overlapping study period, only the study with the largest number of patients was considered in order to avoid duplicated data analysis.

### 2.4. Data Extraction and Quality Assessment

All data from the selected studies were stored in an Excel database which included the following items. For the study design: retrospective vs prospective; for the study population: number of enrolled patients, recruitment period, age, sex, pre-operative pulmonary function; for surgery: type of regional anesthesia, anesthesia time, surgical access, number of biopsies, operative time, need of conversion to thoracotomy and/or general anesthesia; for post-operative outcomes: 30-day mortality, complications, acute exacerbation of ILD, histology, hospital stay. In addition, data from studies, including patients undergoing MVTLB as a control group, were stored in a different database for meta-analysis.

The quality of each article was evaluated according to the QualSyst tool [11] with a summary score varying from 0 to 1.

### 2.5. Data Synthesis and Statistical Analysis

Data were reported as mean ± standard deviation, median (interquartile range), or number (%), unless otherwise specified. Histograms were used to display some categorical data. For a subgroup of included studies with a comparison between SVTLB and MVTLB, we conducted a meta-analysis. The pooled rate was calculated using the Mantel–Haenszel method. The impact of statistical heterogeneity on the pooled estimates of individual outcomes was evaluated using the I^2^ test, which measures the extent of inconsistency among the results of the studies. An I^2^ value greater than 50% indicated significant heterogeneity. The results were graphically displayed using a forest plot. Statistics were performed using RevMan online (Version: 6.4.2).

## 3. Results

### 3.1. Study Search and Selection

The election of eligible studies is summarized in Figure 1. The literature search detected a total of 1601 articles, 47 of which were removed due to being duplicates. After title and abstract screening, 34 studies were selected for full-text review. Twenty of these articles were excluded for the following reasons: two articles were not original research; fifteen articles did not address the study outcomes; two articles could not be located, and two articles met the inclusion criteria but had a partial time overlap with another study from the same author. Fourteen articles were finally selected for the systematic review.

### 3.2. Study Characteristics

The study characteristics are reported in Table 1. 

Thirteen studies for a total of 675 patients (range 10–202) undergoing SVTLB for undetermined ILD were included in the review. Three studies were prospective and 10 were retrospective. The study design was multicentric, bicentric, or single-center in one [14], two [21,23], and ten [10,12,13,15,16,17,18,19,20,22] instances, respectively. Six studies included a comparison between the SVTLB and MVTLB groups [13,15,16,18,19,23]. When looking at the temporal distribution of the published series, two studies were published from 2010 up to 2017 and eleven studies were published from 2018 up to 2023 (Table 1).

With regard to quality assessment, QualSyst scores ranged from 0.50 to 0.95, with a median value of 0.78. None of the studies could be assessable in “blinding” domains. 

### 3.3. Patients’ Characteristics

The age of the enrolled population varied from a minimum mean value of 49.6 years [12] to a maximum median value of 67.4 years old [21], and 358/649 patients (55.2%) were male. In one study, sex was not reported [15]. 

Eleven studies [10,12,13,14,16,17,18,19,20,21,23] reported pre-operative pulmonary function tests, with mean values ranging from 64.6% to 86.3% for forced expiratory volume during the first second, 69.5% to 84.1% for forced vital capacity, and from 51.4% to 68.5% for diffusing capacity for carbon monoxide.

### 3.4. Anesthesia and Surgery

Nine studies [10,13,14,15,16,18,21,22,23] reported details about anesthesia. With regard to regional anesthesia, thoracic epidural analgesia was the chosen technique in three studies [13,16,22], while in the other two investigations [15,23], the enrolled population all received intercostal blocks. In the other studies, both procedures were alternatively performed. 

Neuromuscular blockers were not employed in any study. When applied, sedation was obtained with propofol and/or remifentanil [10,17,18,21,23], propofol and sufentanil [12], remifentanil or dexmedetomidine [13], or midazolam and fenatanil [16]. One study reported that all patients were fully conscious during the procedure [19], while the others did not mention details about sedation [14,15,20,22].

The surgical approach was mentioned in 9/13 studies [10,12,13,14,16,17,20,22,23]. Out of a total of 360 patients, 257 (71.4%) underwent uniportal VATS while 79 (21.9%) underwent multiportal VATS. One multicentric study [14] also reported 24 surgeries (6.7%) performed using a mini-thoracotomy approach under video assistance. Twelve out of thirteen studies reported a mean number of approximately two biopsies for each operation, usually performed through mechanical stapled wedge resections. In one study from Zhang et al. [20], biopsy forceps were employed to obtain the specimens. Conversion to general anesthesia was reported in 5/675 patients (0.7%). Moreover, conversion to thoracotomy occurred in 2/675 cases (0.3%). At the end of surgery, one chest tube was routinely placed with the exception of two studies [12,17], in which a tubeless approach was carried out. Mean or median operative time was reported in 12 studies [10,12,13,14,16,17,18,19,20,21,23] and ranged from a minimum of 22 min to a maximum of 57.7 min.

### 3.5. Histological Results and Diagnostic Yield

The diagnostic yield of surgical biopsies was reported in 12/13 articles. We excluded the study of Hajjar et al. [15] due to a possible selection bias since the authors stated in the material and methods section that all patients had idiopathic pulmonary fibrosis and did not specify if the presence of other diagnoses or undetermined ILD in the histological report was an exclusion criterion. 

A final diagnosis was obtained in 631/649 patients (97.2%). Diagnostic yield ranged from 84.6% to 100% (median diagnostic yield of 100%). In seven studies [15,18,19,20,21,24,25], the enrolled population all received a final diagnosis. The paper with the lowest rate was the one by Zhang et al. [20], in which the biopsies were taken with forceps instead of wedge resections. 

Ten papers [10,12,13,14,16,19,20,21,22,23] mentioned in detail all histological results. These are summarized in Figure 2.

### 3.6. Post-Operative Outcome

All the papers included in the review reported data about post-operative morbidity. Out of a total of 675 patients, 64 (9.5%) had a complication, with morbidity rates ranging from 0.0% [13,16] to 27.3% [22] (median morbidity rate of 7.0%). The complications were usually minor: prolonged air leaks in 15/64 patients (23.4%), pneumonia in 5/64 patients (7.8%), atelectasis in 4/64 patients (6.2%), pain at the tube insertion site in 3/64 cases (4.6%), subcutaneous emphysema in 2/64 cases (3.1%), atrial fibrillation, acute exacerbation of ILD, anemia, gastric bleeding, hematoma requiring revision surgery, thromboembolism, pleural effusion and post-operative tube insertion each occurred in 1/64 patients (1.6%), and other non-specified complications in 27/64 cases (42.1%), 18 of which belonged to Clavien-Dindo grade I [24]. There was no 30-day mortality in the reviewed articles. When reported, hospitalization ranged from a minimum mean value of 1 day [16] to a maximum median value of 8 days [22].

### 3.7. Comparison between SVTLB and MVTLB

Six articles compared SVTLB to MVTLB for ILD [13,15,16,18,19,23]. A total of 178 patients belonged to the former group while 289 patients belonged to the latter. Post-operative complications were recorded in 13/178 patients (7.3%) in the SVTLB group (median morbidity rate of 5.2%) and 71/289 patients (24.6%) in the other group (median morbidity rate of 24.0%). The meta-analysis showed a significantly lower risk of complications (*p* < 0.001) in patients submitted to SVTLB, with an Odds Ratio (OR) of 0.31 (95% confidence interval 0.16–0.59). The studies were homogeneous as I^2^ = 0.0% (Figure 3). With reference to the other main outcome of the review, the comparison of diagnostic yield was performed with five studies. The study of Hajjar et al. [15] was excluded due to selection bias. In addition, no meta-analysis was performed for the diagnostic yield outcome measure since a final diagnosis was reached in all but one case in the MVTLB group [23].

Apart from primary outcomes, there was no difference in mortality (Figure 4). Data were not homogeneous and complete enough to perform a meta-analysis for operative time, anesthesia time, and length of hospital stay. However, three investigations reported significantly shorter mean operative times for the SVTLB group: 32.5 ± 18.5 min versus 50.8 ± 18.4 min (*p* = 0.004) in the study by Kurihara et al. [16]; 38 min versus 77 min (*p* < 0.001) in the study by Guerrera et al. [18]; 27.4 ± 14.4 min versus 36.4 ± 15.1 min (*p* = 0.008) in the study by Grott et al. [23]. Conversely, Jeon et al. [13] reported no significant difference (*p* = 0.25) in operative time, although they found a median value of anesthesia time of 66 min (IQR 62–82) for SVTLB versus 83 min (IQR 69–99) in the MVTLB group (*p* = 0.025). On the other hand, Grott et al. [23] reported a significantly higher anesthesia induction time for the SVTLB group (24.1 ± 15.6 versus 13.9 ± 9.7 min, *p* < 0.001). With reference to the length of hospital stay, significantly shorter times in the SVTLB group were reported by Kurihara et al. [16] (1.0 ± 1.3 versus 10.0 ± 34.7 days, *p* < 0.001), Guerrera et al. [18] (3.1 versus 6.7 days, *p* = 0.0002), and Grott et al. [23] (3.8 ± 1.6 versus 5.7 ± 2.0 days, *p* < 0.001). 

## 4. Discussion

The results of this review have shown that SVTLB is being safely performed in different centers worldwide in patients with undetermined ILD (Figure 5). Although this procedure is still only adopted in high-volume centers, the increasing number of series reported from different countries in recent years suggests an increasing consensus about SVTLB. Moreover, SVTLB was characterized by a highly satisfactory diagnostic yield of nearly 100%, particularly in studies in which the biopsy specimens had been obtained through a mechanically stapled wedge resection. In addition, available data, although still limited, suggest that this novel ultra-mini-invasive surgical approach is associated with lower morbidity than equivalent procedures carried out by MVTLB. 

Surgical biopsy represents the current gold standard for making a histological diagnosis of ILD, as suggested by the latest guidelines [3]. Indeed, this method allows medical professionals to obtain, through mechanically stapled wedge resections, wide specimens including diseased tissue with no architectural distortion, the pleural interface, and some healthy tissue, thus providing the pathologist the most appropriate samples to establish the histologic pattern of the disease. Furthermore, to maximize diagnostic accuracy, it allows biopsies from multiple lobes with different grades of severity of disease to be taken [25]. However, despite the frequent adoption of MVTLB, which has reduced the historically reported, relevant mortality and morbidity rates [26], the procedure is still not considered to be risk-free, with reported mortality and morbidity rates of 2.4–3.6% [6,9] and 7.5–16% [8,27,28], respectively, which are similar to those reported following pulmonary lobectomy [9]. These results demonstrate that surgical biopsy may not be feasible for everyone and careful attention should be paid to the risk–benefit balance, taking into account the stage of disease and patients’ comorbidities. With the aim to reduce the risk of procedure-related complications, SVTLB in patients with undetermined ILDs was proposed by our group as a feasible, less invasive alternative [10].

Importantly, there was no 30-day mortality amongst the studies included in this review. 

This result seems somewhat innovative and promising since, so far, lung biopsy in ILD patients has been associated with a certain operative mortality, which accounted in a recent meta-analysis for 1.7% after MVTLB and 0.6% following transbronchial cryobiopsy [29]. 

As far as morbidity is concerned, according to the Clavien-Dindo classification [24], class 1 complications, not needing any further pharmacological treatment, included prolonged air leaks and atelectasis in 23.4% and 6.2% of patients, respectively, whereas class 2 complications, requiring additional pharmacological therapy, included pneumonia and pain at the tube insertion site in 7.8% and 4.6% of patients, respectively. Moreover, only one case requiring reoperation was reported, which is a rate of 1.6%. 

There are several theoretical reasons underlying the improved results of SVTLB compared to MVTLB. General anesthesia and single-lung mechanical ventilation entail multiple potentially adverse effects triggered by ventilator-associated lung injuries, which include barotrauma, volotrauma, and atelectrauma. Furthermore, the residual muscle block after surgery related to the use of muscle relaxants in MVTLB could cause diaphragmatic dysfunction, weakness of the upper airway muscles, and airway obstruction, leading to hypoxemia [30]. Overall, these adverse effects are more common and dangerous in patients with pre-existing pulmonary impairment, particularly in subjects with low compliance and mechanically heterogeneous lung areas [31], as expected in most patients with ILD. 

The overstretch of low-compliance lung tissue due to positive pressure ventilation and exposure to high oxygen concentrations might also trigger an acute exacerbation of ILD, [32] a life-threatening complication after lung biopsy, defined by Collard as an acute respiratory deterioration with alveolar abnormality in absence of other causes (i.e., drugs or infections) [33]. Furthermore, general anesthesia could also be associated with the induction of cardiac arrhythmia, liver or kidney injury, cognitive dysfunction, or impairment of the immune system. Lastly, double-lumen tube intubation is associated with airway traumatism and can, in rare cases, be complicated by airway laceration [34]. On the other hand, spontaneous ventilation thoracic surgery has been shown to reduce the risk of operative complications and length of hospital stays by preserving spontaneous breathing and assuring more physiologic pressure and volume gradients [35]. These findings, which seem corroborated by the results of this review, hold promise and, if confirmed by future prospective studies, might eventually nominate SVTLB to be considered as the safest and most accurate diagnostic tool for ILD patients. 

As far as diagnostic accuracy is concerned, SVTLB resulted in a highly satisfactory diagnostic yield, which is comparable to the yield reported by MVTLB. A theoretical reason for this result might be that lung and mediastinal movements occurring during spontaneous ventilation might not have affected surgical maneuvering and thus the quality of the specimens. In addition, the finding that both short operative times and high diagnostic yields have been achieved even in series with a small number of cases suggests that for simple procedures such as biopsies for ILD, the learning curve is rapid and satisfactory results may be easily achieved and reproduced. 

Amongst other mini-invasive alternatives to SVTLB, bronchoscopic lung cryobiopsy has shown promising results, by allowing larger specimens to be taken than classical transbronchial biopsy, reaching acceptable diagnostic rates of 76.8–80.7% [29]. Nevertheless, despite the COLDICE study demonstrating a high level of concordance between this technique and surgical biopsy [36], a recent meta-analysis showed lower diagnostic rates for transbronchial cryobiopsy when compared to MVTLB (83.7% versus 92.7%) [5]. Conversely, adequate comparative data between SVTLB and transbronchial cryobiopsy are still lacking. 

In addition, with reference to complications, in the current review, there were no cases of pulmonary bleeding, which is one of the most frequent complications reported with bronchoscopic lung cryobiopsy, occurring in 2.4–22.4% of treated patients, including severe bleeding in 1.6% of instances [29]. 

The null mortality rate found in this review compares favorably with that reported with bronchoscopic lung cryobiopsy, which ranged between 0.3–0.6% in a recent meta-analysis [29,37]. For these reasons, the role of transbronchial cryobiopsy is still a matter of discussion and surgical biopsy is still the recommended procedure whenever feasible [3].

This review has limitations, including the small number of eligible studies for this review and the limited cohorts in most of them. These limits are probably due to the fact that this surgical procedure is still mainly implemented in a few select high-volume centers, although the increasing number of published series in recent years seems to suggest the increasing attention to and adoption of SVTLB worldwide. Lastly, the search was limited only to Pubmed and Google Scholar, which are two of the most relevant scientific databases. 

## 5. Conclusions

Results of this systematic review have shown that SVTLB is safe and associated with lower operative morbidity than MVTLB while assuring similar excellent diagnostic accuracy. Further data from multicentric and randomized studies are warranted to validate these promising results. 

## Figures and Tables

**Figure 1 jcm-13-00374-f001:**
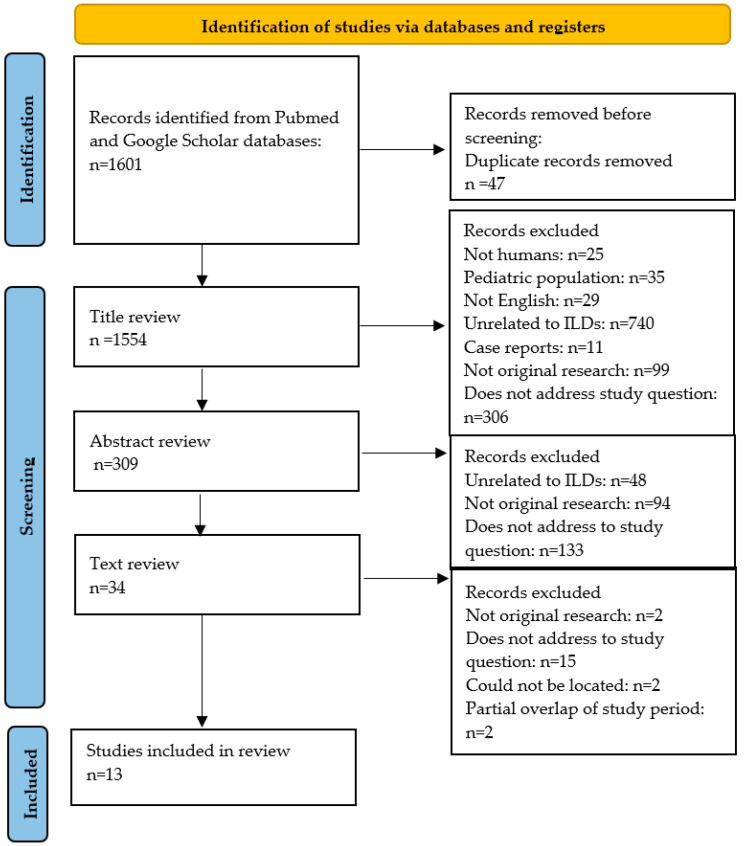
Selection of included studies from the systematic review. ILD: interstitial lung disease.

**Figure 2 jcm-13-00374-f002:**
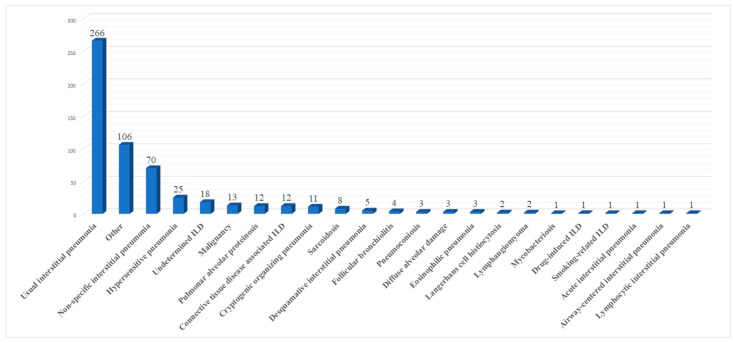
Summary of histological results.

**Figure 3 jcm-13-00374-f003:**
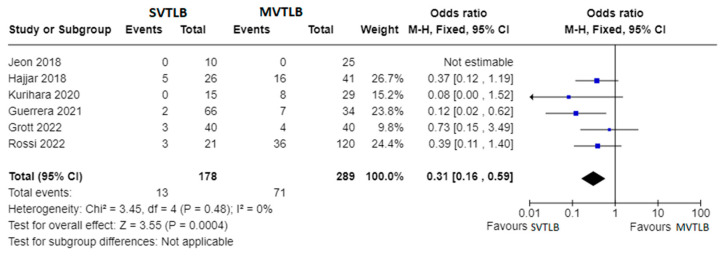
Forest plots of morbidity rate. Squares represent point estimates of the morbidity rate from each study reporting results eligible for analysis. Solid lines represent a 95% confidence interval (CI). The diamond represents the overall pooled effect from the included studies [13,15,16,18,19,23].

**Figure 4 jcm-13-00374-f004:**
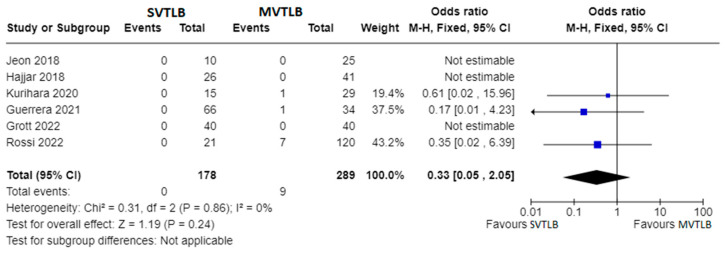
Forest plots of mortality rate. Squares represent point estimates of mortality rate from each study reporting results eligible for analysis. Solid lines represent a 95% confidence interval (CI). The diamond represents the overall pooled effect from the included studies [13,15,16,18,19,23].

**Figure 5 jcm-13-00374-f005:**
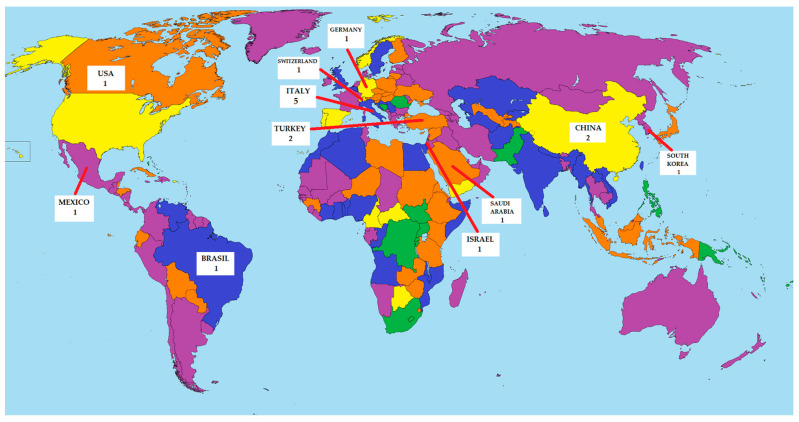
Geographical distribution of studies about SVTLB.

**Table 1 jcm-13-00374-t001:** Main characteristics of the selected studies. MVTLB: mechanical ventilation thoracoscopic lung biopsy; NR: not reported.

First Author, Year	Study Type	Study Period	N. of Patients	Age	Male/Female	Diagnostic Yield	Conversion to Mechanical Ventilation Rate	Post-Operative Complication	Comparison with MVTLB	N. Patient Control Group
Pompeo, 2013 [10]	Prospective	Dec 2009–Jan 2012	30	62.0 ± 10.0	15/15	97.0%	0.0%	1/30 (3.3%)	No	-
Peng, 2017 [12]	Retrospective	Jan 2014–May 2015	43	49.6 ± 10.7	23/20	88.4%	0.0%	3/43 (7.0%)	No	-
Jeon, 2018 [13]	Retrospective	Jan 2016–June 2016	10	61.2 ± 6.6	6/4	100.0%	0.0%	0/10 (0.0%)	Yes	25
Pompeo, 2018 [14]	Retrospective	Jun 2017–Nov 217	112	60.0 ± 12.0	65/47	96.0%	4.5%	8/112 (7.1%)	No	-
Hajjari, 2018 [15]	Retrospective	Jan 2008–Dec 2015	26	NR	NR	NR	0.0%	5/26 (19.2%)	Yes	41
Kurihara, 2020 [16]	Prospective	Mar 2016–Mar 2018	15	62.8 ± 14.7	7/8	100.0%	0.0%	0/15 (0.0%)	Yes	29
Souza, 2021 [17]	Retrospective	Jan 2019–Jan 2020	14	65.8 ^#^	7/7	100.0%	0.0%	2/14 (14.2%)	No	-
Guerrera, 2021 [18]	Prospective	Jun 2016–Feb 2020	66	60.4 ± 2.0	42/24	100.0%	0.0%	2/66 (3.0%)	Yes	34
Rossi, 2022 [19]	Retrospective	Jan 2018–Dec 2020	21	69.5 ^#^	6/15	100.0%	0.0%	3/21 (14.0%)	Yes	120
Zhang, 2022 [20]	Retrospective	Jan 2015–Jul 2021	52	53.6 ± 15.2	25/27	84.6%	0.0%	3/52/5.8%)	No	-
Cherchi, 2022 [21]	Retrospective	Apr 2015–Nov 2021	202	67.4 (60.0–73.5) *	142/60	99.0%	0.0%	22/202 (10.9%)	No	-
Katgi, 2022 [22]	Retrospective	2015–2020	44	56.3 ± 12.6	20/24	100.0%	0.0%	12/44 (27.3%)	No	-
Grott, 2022 [23]	Retrospective	Feb 2013–Apr 2021	40	62.3 ± 10.7	30/10	100.0%	0.0%	3/40 (7.5%)	Yes	40

* values reported as median (interquartile range), # standard deviation was not reported.

## Data Availability

Data are contained within the article.

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
