# Peer review of "Spontaneous Ventilation Thoracoscopic Lung Biopsy in Undetermined Interstitial Lung Disease: Systematic Review and Meta-Analysis"

_jcm, 2024, doi:10.3390/jcm13020374_

Round 1

Reviewer 1 Report

Comments and Suggestions for Authors

Dear authors,

The manuscript you have submitted is well written and contains all the necessary information for clinical use.

In our current practice, we are seeing a rapid decline in indications due to criobiopsy. The only cases we perform are the difficult cases and those that have additional pathology such as a periferal tumour that needs to be resected for diagnosis.

Like you I am a surgeon and I can relate to the written text.

Nevertheless, the COLDICE study, which you correctly cite, shows that the diagnostic yield of transbronchial biopsy is equivalent to that of surgery. I would trust a prospective, randomised, and blinded study much more than a meta-analysis based on lower quality data.

I have no further comments.

Author Response

Thank you for your comment!

Reviewer 2 Report

Comments and Suggestions for Authors

I greatly congratulate the authors for their systematic review, which summarizes the results of spontaneous ventilation thoracoscopic lung biopsy (SVTLB) focusing on diagnostic yield and operative morbidity.

The authors reveal that SVTLB is safe and associated to lower operative morbidity than mechanical ventilation thoracoscopic lung biopsy (MVTLB) while assuring similar excellent diagnostic accuracy.

I strongly agree with the idea that SVTLB is safe and has substantial potential and that further data from multicentric and randomized studies are warranted to validate these promising results of SVTLV.

Therefore, the importance of the study presented is quite high.

 Thank you.

Comments on the Quality of English Language

While I am very enthusiastic about the study they present, I have one minor comment regarding the English of the manuscript. I strongly advise that the authors have a professional English-editing service correct the language in the manuscript, which can then be resubmitted.

Author Response

I greatly congratulate the authors for their systematic review, which summarizes the results of spontaneous ventilation thoracoscopic lung biopsy (SVTLB) focusing on diagnostic yield and operative morbidity.

The authors reveal that SVTLB is safe and associated to lower operative morbidity than mechanical ventilation thoracoscopic lung biopsy (MVTLB) while assuring similar excellent diagnostic accuracy.

I strongly agree with the idea that SVTLB is safe and has substantial potential and that further data from multicentric and randomized studies are warranted to validate these promising results of SVTLV.

Therefore, the importance of the study presented is quite high.

Thank you.

Response: Thank you for your comment!

While I am very enthusiastic about the study they present, I have one minor comment regarding the English of the manuscript. I strongly advise that the authors have a professional English-editing service correct the language in the manuscript, which can then be resubmitted.

Response: We thank the reviewer for the helpful suggestion. The paper has been thoroughly revised by a native English expert. 

Reviewer 3 Report

Comments and Suggestions for Authors

Thank you for the possibility to review the manuscript titled: “Spontaneous Ventilation Thoracoscopic lung biopsy in undetermined Interstitial Lung Disease: Systematic Review and Meta-analysis”. The study is interesting and easy to read. Based on the current results of the meta-analysis SVTLB is safe and associated to lower operative morbidity than MVTLB while assuring similar excellent diagnostic accuracy. The introduction section is well-written. The material and methods section has detailed explanation. The results are consistent with the analysis. There are only several minor recommendations:

-Please review the language of the manuscript. There are several type mistakes;

-Figure 1 has underlined text please change for esthetic reasons;

-“Gender” is a psychological term. Please change it to “sex;

-There are only two databases included in the revision (Pubmed and Google Scholar). This should be included in the limitation or explained that other databases did not have any important articles;

-Figure 5 also demonstrates that this technique is performed is selective centers a limited number of countries. This requires some analysis. The most reasonable explanation is that this procedure was performed only in large centers.

Please take into account the recommendations in the spirit of improving the quality of the submission.

Comments on the Quality of English Language

Minor language editing

Author Response

Thank you for the possibility to review the manuscript titled: “Spontaneous Ventilation Thoracoscopic lung biopsy in undetermined Interstitial Lung Disease: Systematic Review and Meta-analysis”. The study is interesting and easy to read. Based on the current results of the meta-analysis SVTLB is safe and associated to lower operative morbidity than MVTLB while assuring similar excellent diagnostic accuracy. The introduction section is well-written. The material and methods section has detailed explanation. The results are consistent with the analysis. There are only several minor recommendations:

-Please review the language of the manuscript. There are several type mistakes;

-Figure 1 has underlined text please change for esthetic reasons;

Response: Thank you. We changed the Figure as you correctly suggested.

-“Gender” is a psychological term. Please change it to “sex;

Response: Thank you for your suggestion. We changed the term “gender” with “sex” in line 143

-There are only two databases included in the revision (Pubmed and Google Scholar). This should be included in the limitation or explained that other databases did not have any important articles;

Response: We reported this in the limitation as you requested.

-page 11 line 327: “Lastly, the search was limited only to Pubmed and Google Scholar which are two of the most relevant scientific databases anyway”.

-Figure 5 also demonstrates that this technique is performed is selective centers a limited number of countries. This requires some analysis. The most reasonable explanation is that this procedure was performed only in large centers.

Response: We highlighted this element in the discussion as you suggested

-page 8 line 237: “Although this  procedure is still adopted only in few countries in high-volume centres, the increasing number of series reported in recent years seems to suggest an increasing spread of SVTLB”.

-page 11 line 326: “this surgical procedure is still mainly implemented in few selected high-volume centres although the increasing number of published series in recent years seems to suggest an increasing attention and adoption of SVTLB worldwide”.

Please take into account the recommendations in the spirit of improving the quality of the submission.

Response: We greatly thank the reviewer for the constructive criticism and helpful suggestions that we have tried to follow at best.